# Methadone Treatment Gap in Tennessee and How Medication Units Could Bridge the Gap: A Review

**DOI:** 10.3390/pharmacy11050131

**Published:** 2023-08-22

**Authors:** Joanna Risby, Erica Schlesinger, Wesley Geminn, Alina Cernasev

**Affiliations:** 1Tennessee Department of Mental Health and Substance Abuse Services, Andrew Jackson Building, 6th Floor, 500 Deaderick Street, Nashville, TN 37243, USA; erica.schlesinger@tn.gov (E.S.); wesley.geminn@tn.gov (W.G.); 2Department of Clinical Pharmacy and Translational Science, University of Tennessee Health Science Center College of Pharmacy, 301 S. Perimeter Park Drive, Suite 220, Nashville, TN 37211, USA; acernase@uthsc.edu

**Keywords:** opioid treatment program, opioid use disorder, methadone, medication units

## Abstract

The opioid epidemic has been an ongoing public health concern in the United States (US) for the last few decades. The number of overdose deaths involving opioids, hereafter referred to as overdose deaths, has increased yearly since the mid-1990s. One treatment modality for opioid use disorder (OUD) is medication-assisted treatment (MAT). As of 2022, only three pharmacotherapy options have been approved by the Food and Drug Administration (FDA) for treating OUD: buprenorphine, methadone, and naltrexone. Unlike buprenorphine and naltrexone, methadone dispensing and administrating are restricted to opioid treatment programs (OTPs). To date, Tennessee has no medication units, and administration and dispensing of methadone is limited to licensed OTPs. This review details the research process used to develop a policy draft for medication units in Tennessee. This review is comprised of three parts: (1) a rapid review aimed at identifying obstacles and facilitators to OTP access in the US, (2) a descriptive analysis of Tennessee’s geographic availability of OTPs, pharmacies, and federally qualified health centers (FQHCs), and (3) policy mapping of 21 US states’ OTP regulations. In the rapid review, a total of 486 articles were imported into EndNote from PubMed and Embase. After removing 152 duplicates, 357 articles were screened based on their title and abstract. Thus, 34 articles underwent a full-text review to identify articles that addressed the accessibility of methadone treatment for OUD. A total of 18 articles were identified and analyzed. A descriptive analysis of Tennessee’s availability of OTP showed that the state has 22 OTPs. All 22 OTPs were matched to a county and a region based on their address resulting in 15 counties (16%) and all three regions having at least one OTP. A total of 260 FQHCs and 2294 pharmacies are in Tennessee. Each facility was matched to a county based on its address resulting in 70 counties (74%) having at least one FQHC and 94 counties (99%) having at least one pharmacy. As of 31 December 2022, 17 states mentioned medication units in their state-level OTP regulations. Utilizing the regulations for the eleven states with medication units and federal guidelines, a policy draft was created for Tennessee’s medication units.

## 1. Introduction

The opioid epidemic, or the opioid crisis, has been an ongoing public health concern for the last few decades in the United States (US). In this section, the history and current state of the opioid epidemic, opioid use disorder (OUD), methadone maintenance treatment, and opioid treatment programs are reviewed.

### 1.1. Opioid Epidemic

The number of overdose deaths involving opioids, hereafter referred to as overdose deaths, has increased yearly since the mid-1990s [1]. In 2021, there was nearly a 15% increase in overdose deaths from 2020, with almost 90% involving synthetic opioids (i.e., fentanyl) [2]. Totaling over 932,000 deaths since 1999, the epidemic has been ongoing since the 1990s, with the first increase in overdose deaths involving prescription opioids characterizing the first wave of the epidemic [1,3]. The widespread availability and use of prescription opioids during the 1990s are attributed to misleading pharmaceutical marketing, liberal prescribing practices, increased emphasis on pain management, and inadequate drug monitoring [1,3]. Federal and state policies targeting overprescribing, pain management, prescriber education, and prescription monitoring were enacted to reduce the availability and accessibility of prescription opioids [4].

While these efforts have reduced nationwide opioid prescribing by 46.4% in the last decade, overdose deaths continued to increase due to access to heroin in 2010 (the second wave) and illicit synthetic opioids (the third wave), primarily fentanyl and fentanyl analogs, in 2013 [1,3,5]. By 2017, over 15,000 overdose deaths involved heroin, while over 28,000 deaths involved synthetic opioids [2]. That same year, the US Department of Health and Human Services (HHS) declared the opioid crisis a public health emergency, which has been renewed every 90 days for the last six years [3].

### 1.2. Opioid Use Disorder (OUD)

Opioids have a high potential for abuse and psychological and physical dependence because of their ability to induce euphoria in individuals [6,7]. Over time, individuals build a tolerance and require additional opioids to maintain the euphoric effect or to prevent withdrawal symptoms if dependence has developed [6,7]. Opioid dependence is considered a risk factor for OUD [6]. OUD, or opioid addiction, is a chronic condition characterized by the recurrent use of opioids despite significant impairment and distress to one’s life and relationships [6]. The clinical diagnostic criteria set forth by the Diagnostic and Statistical Manual of Mental Disorders (DSM) describes eleven behaviors, including tolerance and withdrawal, associated with compulsive and continual use of opioids despite negative or harmful consequences [8]. As of 2020, approximately 2.7 million individuals in the US have OUD, a 43% increase from 2015 [9].

OUD is a treatable chronic disorder [9]. One treatment modality for OUD is medication-assisted treatment (MAT), a multitherapeutic approach that combines behavioral health interventions with pharmacotherapy [10]. MAT has been associated with a 2.56 risk reduction in all-cause mortality compared to abstinence-only treatment [11]. MAT is further associated with reduced illicit opioid use, a lower risk of overdose, and improved treatment engagement and retention [12,13,14,15]. As of 2022, only three pharmacotherapy options have been approved by the Food and Drug Administration (FDA) for treating OUD: buprenorphine, methadone, and naltrexone [16,17].

### 1.3. Methadone for OUD

Of the three, methadone has the most extensive history and strong evidence for effectiveness and is considered the gold standard compared to other OUD treatments [17]. Methadone is a long-acting full opioid agonist that has been used since the 1960s [17,18]. As a full agonist, methadone can be initiated in individuals without inducing withdrawal and producing cravings [17]. Methadone does not cause euphoria in individuals with OUD because of maintaining and developing opioid tolerance. A comprehensive Cochrane review found that individuals on methadone have 33% fewer unfavorable drug screens and are over four times more likely to remain in treatment even without behavioral interventions and counseling [19]. Methadone may be more effective for individuals misusing shorter-acting opioids such as fentanyl [9]. Methadone has the risk of diversion, but evidence shows that individuals divert methadone for lack of access to medication and that 80% of individuals who divert do so to help others receive care for their OUD. High doses of methadone (60–100 mg/day) are more effective in producing abstinence from illicit opioids than other doses [20]. Compared to buprenorphine and naltrexone, methadone is associated with decreased opioid use, reduced mortality, criminality, and improved patient quality of life (physical, mental, and social well-being) [12]. Unlike buprenorphine and naltrexone, methadone dispensing and administrating are restricted to opioid treatment programs (OTPs), making methadone unique in its accessibility [16].

### 1.4. OTPs and Federal Regulations

OTPs or narcotic treatment programs (NTPs) are the only locations allowed to dispense and administer methadone for OUD. Historically, opioids prescribed for OUD were prohibited by the Harrison Narcotic Act of 1914. It was not until the 1960s, when evidence showing the effectiveness of methadone for OUD compared to short-acting morphine and heroin, was the ban on prescribing opioids to treat OUD was reconsidered [18]. In 1972, the FDA regulated the use of methadone for OUD. In 1974, the Narcotic Addict Treatment Act (NATA) was passed, amending the Controlled Substance Act to allow for maintenance treatment for OUD. Federal regulations for OTPs were issued and implemented to govern the use of methadone for OUD and shift administration and oversight to the Substance Abuse and Mental Health Services (SAMHSA) [18]. Additionally, states can regulate OTPs on a state level creating a multi-layered regulatory system that varies from state to state [21].

Federal regulations for OTPs, 42 Code of Federal Regulations (CFR) Part 8, detail the minimum requirements for OTP licensure, operations, and services, including medication management (Table 1) [22]. SAMHSA serves as the federal authority that oversees, enforces, and provides guidance for OTPs [23]. Under 42 CFR Part 8, existing OTPs are allowed to establish medication units. Medication units are defined as geographically separate facilities under an existing OTP that administer or dispense an opioid agonist treatment, primarily methadone, via a licensed provider or pharmacist [22]. SAMHSA states that medication units are allowed to perform all OTP services if space and staffing permit for quality patient care. For services unable to be provided at the medication unit, those services must be conducted at an OTP. While medication units are subjected to the same regulations and guidelines as OTPs, they offer individuals an alternative patient care site to receive OUD treatment services [23].

### 1.5. Aim

The opioid crisis has contributed to a growing need for MAT services as the prevalence of OUD continues to rise. In Tennessee, opioid-related deaths increased by 54% from 2019 to 2020, with at least 2.4% of Tennesseans affected by OUD [24,25]. Given the presence of fentanyl in almost 90% of Tennessee’s overdose deaths, there is an increased demand for OUD medications to assist treatment and recovery efforts [24]. Medication units offer a unique solution by establishing additional locations where individuals can receive OTP services, including methadone dispensing with minimal additional infrastructure, especially if established in a location with its own per existing DEA number (i.e., federally qualified health centers and pharmacies) [23]. To date, Tennessee has no medication units, and administration and dispensing of methadone are limited to licensed OTPs. This study details the research process used to develop a policy draft for medication units in Tennessee.

This research involves identifying US states with regulations governing medication units, assessing Tennessee’s need for increased geographical availability of methadone dispensing, and identifying obstacles and facilitators to methadone access in the US. This paper used OTPs to symbolize methadone dispensing unless otherwise stated.

The results will present the current and potential geographical availability of methadone dispensing in Tennessee and obstacles and facilitators to methadone access in the US, then conclude with a discussion on any foreseeable barriers to utilizing medication units and the potential impact medication units will have on Tennessee’s geographical availability of methadone dispensing.

## 2. Methods

The methodology detailed below is composed of a rapid review aimed at identifying obstacles and facilitators to OTP access in the US, a descriptive analysis of Tennessee’s geographic availability of OTPs, pharmacies, and federally qualified health centers (FQHCs), and policy mapping of 21 US states’ OTP regulations. This methodology uses OTPs to represent methadone dispensing due to federal laws restricting dispensing and administering methadone for OUD to only OTPs and its medication units. Access refers to an individual’s ability to engage in care, and availability refers to the presence of a facility.

### 2.1. Rapid Review

A rapid review approach was used to synthesize existing literature within a six-week timeframe by streamlining or omitting components of a traditional systematic review. This rapid review followed Cochrane Rapid Review Interim Guidance and was conducted to identify obstacles and facilitators to OTP access in the US [26].

#### Study Selection, Screening, and Inclusion and Exclusion Criteria

Two electronic databases, PubMed and Embase, were searched. A broad search strategy was used to identify peer-reviewed articles relevant to the accessibility of methadone and opioid treatment programs for adults in the United States. The search was limited to English-language articles published before 31 October 2022. Medical Subject Headings (MeSH) terms and keywords with Boolean and truncation operators were used to capture terminology variations and ensure comprehensive search results. MeSH terms used were “methadone”, “opioid-related disorders”, “opiate substitution treatment”, and “health services accessibility”. Keywords used were “opioid treatment program”, “opioid/opiate”, “access”, “availability”, and “opioid use disorder”.

Results from the initial database searches were imported to EndNote. After initial importation, duplicate records were removed, and the remaining abstracts were reviewed for preliminary eligibility determination. After the preliminary review, a full-text review of the remaining articles was conducted to determine their eligibility for the rapid review. The references list for eligible articles was visually scanned to identify additional relevant studies that were not captured in the initial database searches.

Eligible studies were included if they addressed factors influencing OTP enrollment, engagement, and retention. Articles were excluded if the population was outside the US, institutionalized (in-patient, incarcerated), adolescent, or pregnant. Articles focused on buprenorphine, naltrexone, abstinence-based treatment, infectious disease, pain management, and harm reduction were excluded to ensure that the evidence primarily pertained to methadone treatment for OUD.

### 2.2. Descriptive Analysis

A descriptive analysis was used to model the current and potential geographic availability of methadone dispensing in Tennessee on a county and regional level. Geographic availability was examined relative to treatment need and rurality. Treatment need was based on Tennessee’s nonfatal overdoses involving opioid rates. Availability was based on the presence or absence of a facility (i.e., OTP, FQHC, or pharmacy). Current availability was based on licensed OTPs in Tennessee, and potential availability was based on the cumulative number of licensed OTPs, FQHCs, and pharmacies.

#### Data Sources

Geographical locations of all currently licensed OTPs in Tennessee were obtained from the Tennessee Department of Mental Health and Substance Abuse Services (TDMHSAS) Tennessee Opioid Treatment Clinic Map [27]. TDMHSAS is responsible for licensure and administrative oversight of state OTPs [28]. The map was updated on 4 April 2022, and the state has yet to license any new facilities as of March 2023 [27]. Geographic locations of active Tennessee pharmacy licenses were obtained for the Tennessee Department of Health licensure search [29]. Pharmacies located outside of the state of Tennessee were excluded from the analysis. Geographic locations of FQHCs in Tennessee were obtained from the Health Resources and Services Administration (HRSA) Federally Qualified Health Centers and Look-Alikes by State Report [30]. For this study, FQHC designations and Look-Alike designations were not distinguished between.

Nonfatal overdose rates for 2021 at the state and county level were extracted from the Tennessee Drug Overdose Dashboard [24]. The Dashboard is developed and maintained by the Tennessee Department of Health Office of Informatics and Analytics. Nonfatal overdose data is sourced from the Tennessee Department of Health Hospital Discharge Data System and yearly population data for rate calculations from the CDC Wonder. Rates were age-adjusted per 100,000 Tennesseans. Rates were suppressed if the number of nonfatal overdose cases was less than 10.

The geographic designation (urban versus rural) of each county was based on the US Department of Agriculture (USDA) 2013 Rural-Urban Continuum Codes (RUCCs) [31]. The RUCCs classify counties by population size and degree of urbanization and use data from the US Census Bureau 2010 Decennial Census [32]. Codes 1–3 represent metropolitan counties, and 4–9 represent nonmetropolitan counties. This study categorized codes 1–5 as urban and 6–9 as rural counties. The 2013 RUCC was updated in December 2020, and the 2023 RUCC is scheduled to be released in mid-2023 [31]. Population data at the state and county level were obtained from the U.S. Census Bureau 2010 Decennial Census [32].

The primary outcome, the availability of a facility within a county, was dichotomized (yes/no). Nonfatal overdose data, urban-rural designation, and population data were linked to facility data by county name. Availability of dispensing was calculated as a ratio between facility number (total number of OTPS, FQHCs, and pharmacies) to population which is one of the measures the US Department of Health and Human Services (HHS) uses to designate areas with a shortage of healthcare services.

### 2.3. Policy Mapping

Utilizing the public health law research mapping framework, OTP regulations for 21 US states were systematically analyzed to identify common regulations and assess Tennessee’s OTP regulations for parity [33].

The US states included in this analysis are either contiguous with Tennessee’s state border or specifically mention medication units in their OTP regulations.

State-level OTP regulations were identified using a keyword search on Westlaw Legal Research Database. Search terms were “opioid treatment program” and “narcotic treatment program”. Westlaw is an electronic database utilized in legal research to identify state and federal regulations, statutes, and case law.

Regulations were excluded if repealed, not related to the treatment of OUD, general licensure requirements for practitioners, specific to buprenorphine or naltrexone, creating or regulating task forces or committees, general prescribing, administration, and dispensing of controlled substances. Regulations about services for methadone dispensing and administration were the primary focus of this search. Regulations about behavioral health, counseling, and social services were excluded.

The primary outcome was the presence or absence of regulations within a given state. Information was extracted from contiguous states to establish standard regulatory practices, and then states with medication units were examined for new or additional regulatory rules.

## 3. Results

### 3.1. Rapid Review

A total of 486 articles were imported into EndNote from PubMed and Embase. After removing 152 duplicates, 357 articles were screened based on their title and abstract. Studies conducted outside the US (*n* = 112) and hospitalized (*n* = 4) were excluded. Articles on patients that are considered special populations (i.e., adolescents [*n* = 3], geriatric [*n* = 1], incarcerated (*n* = 13), pregnant [*n* = 28], and veterans [*n* = 6]) were excluded due to differences in standards of care for these populations. Articles primarily focused on buprenorphine (*n* = 37) or naltrexone (*n* = 1) were excluded, along with articles on harm reduction (*n* = 35), behavioral health (*n* = 21), infectious disease (*n* = 25), and pain management services (*n* = 8). Lastly, articles reviewing OUD, MOUD, or MAT (*n* = 14) were excluded due to a lack of specific interventions and outcomes.

The remaining 34 articles underwent a full-text review to identify articles that addressed the accessibility of methadone treatment for OUD. A total of 18 articles were identified, three of which were not captured in the database search, to address either a barrier or facilitator of methadone treatment.

#### Summary of Results

The 18 articles comprised in this rapid review present barriers and facilitators to OTP access in the US.

These studies highlight obstacles that hinder methadone access and present the primary advantages of using this treatment. These factors influence treatment access by impacting adherence, retention, or engagement. These articles are categorized as geographic, socioeconomic, or policy, and their impact on treatment access was documented.

Geographic barriers were identified as rural location, travel time, and distance. For example, Amiri et al. found that distance poses a challenge for clients living more than 10 miles away from the OTP [34,35]. This study demonstrated that clients were 44% more likely to miss a dose in the first 30 days of treatment and had a 7% higher dropout rate with each mile increase in distance from the OTP in the first 90 days [34]. Similarly, Joudrey et al. reported that the mean drive time in urban counties is less than eight minutes compared to at least 49 min in rural counties [36]. Furthermore, Bonifonte et al. estimated that 18.2% still need access to an OTP within a 25-mile drive, resulting in an unmet demand for the treatment [37]. This study highlighted that opening a new OTP can decrease travel distance from 11.9 to 6.3 miles/person/day if optimal located [37]. In the same vein, having access to a pharmacy could remove some of the obstacles. For instance, a study determined that the median drive time from a rural population center to the nearest chain pharmacy is 13.3 min compared to 48.4 min to the nearest OTP [38]. The importance of expanding methadone administration beyond OTPs into other facilities, such as FQHCs or pharmacies, will make treatment accessible within a 30 min drive from all population centers was concluded in another study [39].

A recent scoping review suggests that office-based methadone and pharmacy dispensing can improve methadone access for treatment-stable clients while increasing treatment satisfaction and quality of life [40]. Brooner et al. and Wu et al. both conducted pilot studies in independent community pharmacies to demonstrate the feasibility of pharmacy-based methadone dispensing in the US [41,42]. Collaborating with OTP treatment providers, each pharmacy dispensed methadone to treatment-stable patients once every two weeks [41,42]. Each study reported high treatment adherence, care satisfaction, and minimal methadone diversion [41,42].

Socioeconomic barriers were identified as unemployment, insurance coverage, and socioeconomic status. Amiri et al. observed that low attendance for clients living less than five miles from OTP coincided with low socioeconomic status [34]. Another study found that self-paying clients experience shorter admission delays than insured clients [43]. A survey conducted in Michigan identified nine barriers to care, including employment schedules, childcare responsibilities, housing instability, lack of transportation, legal obligations, and treatment costs [44]. The study found that 68.9% of clients have at least one identified retention barrier, and 53.6% have multiple barriers [44].

Three articles referenced take-home privileges as a factor influencing treatment access. The utilization and impact of relaxed regulations for methadone take-homes during the COVID-19 pandemic were examined by Figgatt et al., Levander et al., and Hoffman et al. [45,46,47]. Two qualitative studies found that the number of take-home doses for clients in treatment for at least 180 days increased with little reported diversion, change in drug screen results, and retention rates [45]. A qualitative study by Hoffman et al. highlighted the themes that reflect the clients’ perspectives on receiving increased take-home doses: increased trustworthiness, reduced travel time leading to increased employment and recreational time, and reduced exposure to potential triggers and stigma [47]. In another qualitative study by Levander et al., clients echoed these advantages as clients reported three benefits of increased take-home doses: enhanced self-esteem and feelings of normalcy, reinforcing and supportive of recovery, and reclaiming time spent traveling doing other rewarding activities [46].

The COVID-19 pandemic has influenced the treatment capacity and time to admission, causing delays in care. A study conducted between May and June 2020 highlighted that new admissions were halted at 40 OTPs due to COVID-19 concerns and that the median time to first appointment ranged from 3 to 4 days [48]. Similarly, Madden et al. demonstrated that implementing an open-access model increased patient census by 183% and reduced the time to the first appointment from almost eight days to less than one day in nine years [49]. Thus, McCarthy et al. suggested that interim methadone dosing could facilitate methadone for clients waiting to be admitted into an OTP [50].

### 3.2. Descriptive Analysis

Tennessee has 22 OTPs, 95 counties, and three regions (Eastern, Western, and Middle). All 22 OTPs were matched to a county and a region based on their address, resulting in 15 counties (16%) and all three regions having at least one OTP. A total of 260 FQHCs and 2294 pharmacies are in Tennessee. Each facility was matched to a county based on its address, resulting in 70 counties (74%) having at least one FQHC and 94 counties (99%) having at least one pharmacy.

An OTP was four times more likely to be in an urban county than a rural one. Eastern Tennessee has seven OTPs across six counties (one rural and five urban). Middle Tennessee has seven OTPs across four counties, all urban counties. Western Tennessee has eight OTPs across five counties (two rural and three urban).

Nonfatal overdose rates involving opioids were used to represent treatment needs because overdose is an opioid-related harm, and a nonfatal overdose (hereafter referred to as overdose) indicates a living individual that could potentially seek treatment. Counties with a higher overdose rate than the state’s overdose rate were labeled as counties with a high need. Counties with an overdose rate equal to or lower than the state’s rate were labeled as counties with a low need. Overdose rates were suppressed in 24 counties due to fewer overdoses than 10. Geographic availability of treatment or treatment availability was based on the presence of a facility within a county or its bordering counties to account for cross-county travel.

In 2021, Tennessee’s nonfatal overdose rate was 64 per 100,000 residents. The county with the highest overdose rate at 244 overdoses per 100,000 Tennesseans was Cheatham, an urban county in Middle Tennessee. The county with the lowest overdose rate at 20 overdoses per 100,000 Tennesseans was Hawkins, an urban county in Eastern Tennessee. On average, rural counties had a lower overdose rate than urban counties (83 vs. 97), and Middle Tennessee had a higher overdose rate than Western and Eastern Tennessee (102 vs. 81 vs. 83).

A total of 55 high-need counties were identified. The regional distribution of the counties is as follows: 21 counties are located in Eastern Tennessee, 27 counties are located in Middle Tennessee, and 7 counties are located in Western Tennessee. Treatment was available in 42 of these high-need counties (76%). The remaining 13 high-need counties lacked treatment availability and were majority rural (61%) and located in either Eastern (31%) or Middle Tennessee (69%). A total of 16 low-need counties were identified, with four counties (25%) lacking treatment availability. All four counties were rural counties in Middle Tennessee except Scott, a rural county in Eastern Tennessee.

### 3.3. Policy Mapping

As of 31 December 2022, 17 states mentioned medication units in their state-level OTP regulations. Three states (KY, MO, VA) are contiguous with Tennessee. One state (PA) has banned medication units. Two states (ME, VA) refer to medication units without a formal definition. Three states (IA, ND, KY) provide a formal definition only. The remaining states (CA, FL, MA, MO, NV, OH, OK, OR, SC, TX, WI) had regulations governing medication units. States were compared based on take-home schedule and take-home eligibility plus personnel and licensure requirements (Table 2).

## 4. Discussion

### 4.1. Rapid Review

The rapid review of the literature identified three types of barriers and facilitators to OTP access: geographic, socioeconomic, and policy related. Roughly half of the literature in this review focused on the role of geographic access and availability of OTPs, with the common outcome or variable being travel time or distance. Significant travel times and distances were associated with less frequent treatment engagement and lower medication adherence. Bonifonte, Iloglu, and Joudrey propose that additional methadone dispensing facilities, either as OTPs or medication units in the form of pharmacies or FQHCs, can reduce the travel burden on clients [36,37,39].

Brooner and Wu demonstrated the feasibility of pharmacy-based methadone dispensing with an independent community pharmacy in North Carolina and two pharmacies, one a hospital outpatient and the other an independent community, in Maryland [41,42]. Both pilot studies had high satisfaction rates with providers and clients. In each study, clients went to the pharmacy weekly or biweekly for an observed dose and to pick up six or thirteen take-home doses. Clients were only eligible if their current take-home schedule at the clinic was reflective of these schedules. In Maryland, clients were only eligible for at least six take-home doses after nine months of continuous treatment. In North Carolina, clients were only eligible after one year of ongoing treatment.

Future studies should address patients who have been in treatment for less than nine months due to the high rate of treatment dropout and nonadherence among clients in early treatment, especially those whose primary reason for dropout and nonadherence was travel distance or time. A potential barrier to pharmacy dispensing not found in the rapid literature review is pharmacies and pharmacists’ willingness to engage in pharmacy-based methadone dispensing. Previous research on pharmacies and pharmacists dispensing buprenorphine for OUD suggests stigma and lack of education will be barriers [51].

SAMHSA, the federal authority that dictates the maximum allowable take-home doses, increased the flexibility of their take-home schedule to increase access to take-home quantities for qualifying clients during the pandemic [52]. SAMHSA’s proposed rule change in 42 CFR Part 8 in December 2022 has refined and clarified these exceptions to make the flexibilities permanent. If the proposed rule is adopted into legislation, clients could engage in a weekly take-home schedule, such as the schedule utilized in the Wu study, regardless of time in treatment. While accelerating access by establishing take-home flexibility to increase patient access, diversion is always a concern. Three qualitative studies examining the impact of the change in take-home doses during COVID-19 reported little to no diversion [44,45,47]. These studies highlighted patient-identified benefits of increased take homes resulting in a high program and treatment satisfaction rate, which are positively associated with treatment retention. Additional studies should be conducted on the impact of the SAMHSA COVID-19 Exception Wavier for take-home doses on retention and adherence rates.

Due to cumulative geographic and socioeconomic barriers identified in the rapid review, rural areas need more access to OTPs. The rapid review brought forward the office-based dispensing of methadone through FQHCs. FQHCs provide care to 9% of the US population, including one in five rural Americans, making them well-positioned to expand methadone access in these areas [53]. FQHCs are currently being used in Ohio to expand methadone access [54].

Additionally, services provided by FQHCs must be covered by state Medicaid programs [55]. While cost was not explicitly identified in the rapid review, one study suggested that insurance coverage (self-paying versus insurance) can delay admission and affect the cost of care [48]. Further research on insurance coverage and OUD treatment services provided literature on Medicaid expansion programs covering OUD services. Medicaid expansion to cover OUD services has demonstrated increased enrollment in OUD programs, especially among individuals with Medicaid coverage [56]. A cost analysis conducted in Vermont further showed that Medicaid expansion could reduce the economic burden of OUD on the state’s economy [57]. More research is needed to be performed on the impact of insurance coverage for methadone on opioid-related health outcomes. Some research suggests that the effects of insurance expansion cannot be accurately assessed due to limited provider capacity.

### 4.2. Descriptive Analysis

The descriptive analysis demonstrated a methadone treatment gap in Tennessee. To the authors, this is the first time Tennessee’s OTP availability has been examined on a county, regional, and state level. Tennessee has OTPs within 15 counties, making in-county methadone dispensing available for 16% of the state, which represents 54% of the state’s population. Counties without an OTP in their county or neighboring their county were primarily in East and Middle Tennessee and rural. The current availability of OTPs per 100,000 persons in Tennessee is almost three times less than the national average [58]. Most of Tennessee’s OTPs are operating at 80% capacity at least and with a 25% increase in methadone utilization, which may limit methadone access as facilities reach capacity [59,60]. Medication units offer a way to expand methadone availability without opening a new OTP. Medication units are being utilized in several states to help expand access. Standard medication units used throughout the US are FQHCs and pharmacies. If methadone dispensing were developed for FQHCs in Tennessee, the number of counties without methadone access would decrease by 71%. With the addition of pharmacy dispensing, every county has access to methadone dispensing.

Previous studies have identified that methadone treatment gaps exist across the US, and a few states have examined the treatment gap in their states [61]. Georgia and Ohio have modeled how adding FQHCs-based dispensing can improve access to methadone [62]. Joudrey modeled how pharmacy-based dispensing can improve methadone access in Indiana, Kentucky, Ohio, Virginia, and West Virginia [36]. While medication units have been implemented in some of these states, data on their utilization and impact on methadone access have yet to be reported [63].

### 4.3. Policy Mapping

This review compared OTP regulations from 21 US states to assist in developing a policy draft to govern medication units in Tennessee. Beyond a formal definition, 12 states have regulations governing medications. One state, PA, bans medication units. Of the remaining eleven states, only Missouri borders Tennessee. Compared to its other bordering states, Tennessee’s regulations are uniquely designed to facilitate methadone access via medication units. For example, Tennessee’s three southern contiguous states require that a pharmacist be involved in the methadone administration and dispensing process. Tennessee does not have this additional personnel requirement allowing medication units to be staffed solely by a mid-level practitioner. Additionally, Tennessee’s take-home schedule offers an advantage compared to bordering states with stricter schedules by allowing more patients to utilize the medication unit earlier in treatment. Lastly, Tennessee’s soft cap on methadone dosing and mandatory review of the prescription drug monitoring database help reduce the risk of methadone toxicity and drug interactions.

Of the states with medication unit regulations, several had similar rules to Tennessee. For example, South Carolina requires a certificate of need to establish an OTP or a medication unit. Both California and Iowa have high dose restrictions in place. On the other hand, some state regulations could hinder access to methadone, such as the dosing restrictions in Texas or the lack of take-home doses in the first 30 days of treatment in Wisconsin, regardless of OTP closures.

Utilizing the regulations for the eleven states with medication units and federal guidelines as a reference, a policy draft was created for Tennessee’s medication units. The draft comprises three sections: medication unit establishment, patient eligibility criteria and referral process, and record-keeping requirements. This policy was drafted to encourage current OTPs in Tennessee to establish medication units to expand methadone dispensing. Previous research has demonstrated a policy’s ability to influence the availability and effectiveness of OTP services, which can explain the current variations in treatment utilization and health outcomes across different US states [21,64]. Given the multi-layered regulations governing OTPs, future research should include a thematic analysis of state-level regulations to help identify and isolate regulations and their impact on health outcomes.

## 5. Limitations

### 5.1. Rapid Review

The rapid review resulted in a limited number of qualitative studies conducted in the US. The review only included studies in the English language; literature in other languages was missed. Additionally, grey literature was not included, and some publications could have been missed because they might not have been correctly indexed in the database at the time of the search.

### 5.2. Descriptive Analysis

The descriptive analysis focused on the availability of methadone dispensing in Tennessee rather than accessibility. Availability does not equate to accessibility. Additionally, the availability of facilities was calculated using a ratio that did not account for geographical distance, travel distance, travel time, or an individual’s ability to utilize care in different or multiple regions. As a result, this analysis may oversimplify the availability of these facilities. Lastly, nonfatal overdose rates may overestimate treatment needs.

## 6. Conclusions

The opioid crisis has highlighted a treatment gap for OUD. This review seeks to contribute to the policy and practice of OUD by developing a policy for establishing and operating medication units in Tennessee that can be implemented in the future.

## Figures and Tables

**Table 1 pharmacy-11-00131-t001:** Minimal federal requirements for OTPs.

Minimum Federal Requirements for OTPs		
Regulatory Area	Federal Requirements	
Certification	Application: Accreditation Status, Organizational Structure, Name of Responsible Parties, Facility’s Address (Include Medication Units), Sources of Funding, and Statement of Compliance	
	Certification Renewal Period: Every 3 Years	
Personnel	Program Sponsor, Medical Director, and Counselor	
Required Services	Initial Full Medical Exam within 14 Days of Admission	
	Initial and Periodic Assessments of Services (Medical, Social, Psychological, Educational, Vocational, and Employment)	
	Counseling	
	Medication Monitoring—Eight Random Drug Screens Annually	
Medication Administration, Dispensing, and Use	Only Dispense Oral Formulations of Methadone	
	Dispensing or Administration Limited to Healthcare Professionals Authorized to Administer or Dispense Opioids (i.e., Pharmacists and Registered Nurses)	
	Maximum Initial Dose: 30 mg; Maximum Total Dose on Day 1: 40 mg	
Take-Home or “Unsupervised” Medication Doses	All Patients Allowed One Take-Home Dose for a Scheduled Closure (i.e., Sundays or Holiday)	
	Eligibility: Absence of Illicit Use, Regular Attendance, No Behavioral Problems, No Criminal Activity, Stable Home and Social Environments, Safe Storage Capability, and Benefit Outweigh Risk	
	Schedule	First 90 Days: One Dose Weekly Plus Closure Dose
		Second 90 Days: Two Doses Weekly Plus Closure Dose
		Third 90 Days: Three Doses Weekly Plus Closure Dose
		Fourth 90 Days: Max Six Doses Weekly
		After a Year: Max Two Weeks
		After 2 Years: Max Monthly

**Table 2 pharmacy-11-00131-t002:** Common state OTP regulations.

Common State Minimum Regulations		
Regulatory Area	Regulations	
Certification	Certificate of Need (CON) Required	
	Zoning Restrictions (i.e., Maximum Number of Facilities in a Region, Minimum Distance Requirements Between Facilities)	
	Subject to Pharmacy Licensure	
Personnel	Licensed Pharmacist Required	
	Maximum Patient to Staff Ratios	
	Mid-Level Practitioner Required	
Required Services	Minimum Counseling Requirements	
	Utilization of the Prescription Drug Monitoring Database Required	
	Regulate Hours of Operation (i.e., Minimum Days/Hours Open)	
Medication Administration, Dispensing, and Use	Maximum Maintenance Dose Restrictions/Discouragement	
Take-Home or “Unsupervised” Medication Doses	No Additional Take-Homes Allowed for Closures	
	Additional Eligibility Requirements	
	Schedule	No Take-Home Doses in First 30 Days
		No Take-Home Doses in First 90 Days
		No Take-Home Doses for “High Dose”

## Data Availability

Not applicable.

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
