# Peer review of "Methadone Treatment Gap in Tennessee and How Medication Units Could Bridge the Gap: A Review"

_pharmacy, 2023, doi:10.3390/pharmacy11050131_

Round 1

Reviewer 1 Report

This is an interesting and useful study about how access to medication assisted treatment for OUD is distributed in one state and what might be one policy change that would allow additional medication units to be created.  It is a worthwhile study but needs to be improved in a few ways in order to be a stronger manuscript. 

First, there is no clear definition of a 'medication unit' in the manuscript.  Some general information is provided in section 1.4.  However, the details of the different ways that other states or entities have experimented with FQHCs or pharmacies for dispensing of methadone only come out late in the manuscript, including the guidelines for length of prior treatment required for use, the take-home and dosing limitations.  Those should all be included in a concise definition of what a medication unit is at the beginning of the manuscript. 

Second, there is reference to the three types of medication assisted treatment but no discussion as to why only methadone dispensing is the focus of this discussion when buprenorphine, and naltrexone can also be used, and in some cases have addressed some of the barriers in distribution of care that seem to be a focus of this study. 

Third, the approach to describing the geographic distribution of available OTP, FQHCs and pharmacies with regard to counties and calculations of overdoses per 100,000 population is useful.  However at times the authors seem to state that rural counties are less well served, while they also report that 65% of high need counties are urban; and low need counties based on median overdose rates/100,000  still have a 25% unmet need.  This part of the data reporting just needs to be clarified. What does 'meeting the need for treatment' mean?  see line 346.  Just that there is one OTP in a county???  At the end of the manuscript (line  420 ) this is changed to say 82% of counties have OTP access in at least a neighboring county.  This distinction should be reported earlier in the manuscript when the descriptive data are first presented and with more clarity about what 'unmet need' means if it is not based on each county and rates of overdose per 100,000.  The distribution of overdose rates and the distribution of OTPs seem to be confused as to which show need/unmet need.

Fourth, the Discussion section is also confusing since it doesn't line up with the 3 areas of focus of the study.  It seems new information is being introduced here that wasn't reported earlier.  Also the discussion of insurance (lines 406-415) is confusing.  Lack of insurance is earlier defined as a barrier to care, but here self-pay vs insurance is noted as increasing timeliness of care.  Which is it?  Is the issue between private insurance (where preauthorization might have to occur) and public insurance???

A few minor sentence structure issues: 

line 343 lacks subject-verb agreement

line 435 refers to 'adding either FQHC based dispensing..." without a modifier for the word 'either',  Do the authors mean to also refer to pharmacies here?

Should line 443 say 'medication units' and not just 'medications'.

Line 454 refers to Tennessee having similar medication unit regulations.  The development of medication units and accompanying regulations in Tennessee seems to be the aim of this study, so this is confusing.  It appears that the authors mean that there are other regulations regarding OUD that set up fewer barriers to developing these new services but this needs to be clarified.

English language is fine except for a few awkward sentences.

Author Response

This is an interesting and useful study about how access to medication assisted treatment for OUD is distributed in one state and what might be one policy change that would allow additional medication units to be created.  It is a worthwhile study but needs to be improved in a few ways in order to be a stronger manuscript. 

First, there is no clear definition of a 'medication unit' in the manuscript.  Some general information is provided in section 1.4.  However, the details of the different ways that other states or entities have experimented with FQHCs or pharmacies for dispensing of methadone only come out late in the manuscript, including the guidelines for length of prior treatment required for use, the take-home and dosing limitations.  Those should all be included in a concise definition of what a medication unit is at the beginning of the manuscript. 

Response: Thank you for suggesting adding a “medication unit” in this manuscript. We revised the manuscript and added a definition in the introduction.

Second, there is reference to the three types of medication assisted treatment but no discussion as to why only methadone dispensing is the focus of this discussion when buprenorphine, and naltrexone can also be used, and in some cases have addressed some of the barriers in distribution of care that seem to be a focus of this study. 

Response: Thank you for this suggestion. However, methadone is the main focus of this study due to the most restricted in access.

Third, the approach to describing the geographic distribution of available OTP, FQHCs and pharmacies with regard to counties and calculations of overdoses per 100,000 population is useful.  However at times the authors seem to state that rural counties are less well served, while they also report that 65% of high need counties are urban; and low need counties based on median overdose rates/100,000  still have a 25% unmet need.  This part of the data reporting just needs to be clarified. What does 'meeting the need for treatment' mean?  see line 346.  Just that there is one OTP in a county???  At the end of the manuscript (line  420 ) this is changed to say 82% of counties have OTP access in at least a neighboring county.  This distinction should be reported earlier in the manuscript when the descriptive data are first presented and with more clarity about what 'unmet need' means if it is not based on each county and rates of overdose per 100,000.  The distribution of overdose rates and the distribution of OTPs seem to be confused as to which show need/unmet need.

Response: Thank you for this recommendation. The manuscript has been amended and revised the language around unmet need to treatment availability.

Fourth, the Discussion section is also confusing since it doesn't line up with the 3 areas of focus of the study.  It seems new information is being introduced here that wasn't reported earlier.  Also the discussion of insurance (lines 406-415) is confusing.  Lack of insurance is earlier defined as a barrier to care, but here self-pay vs insurance is noted as increasing timeliness of care.  Which is it?  Is the issue between private insurance (where preauthorization might have to occur) and public insurance???

Response: Thank you for this valuable recommendation. The manuscript was revised and additional information was added. For example, a new paragraph was added that identified the insurance coverage  as a barrier in the rapid review. The manuscript also highlighted further research that  bring more evidence pertaining to Medicaid expansion’s impact on use of OUD services to help strengthen the evidence to support the insurance coverage barrier.

A few minor sentence structure issues: 

line 343 lacks subject-verb agreement

line 435 refers to 'adding either FQHC based dispensing..." without a modifier for the word 'either',  Do the authors mean to also refer to pharmacies here?

Response: Thank you for this suggestion. We amended the text.

Should line 443 say 'medication units' and not just 'medications'.

Response: Thank you for this suggestion. The text was revised.

Line 454 refers to Tennessee having similar medication unit regulations.  The development of medication units and accompanying regulations in Tennessee seems to be the aim of this study, so this is confusing.  It appears that the authors mean that there are other regulations regarding OUD that set up fewer barriers to developing these new services but this needs to be clarified

Response: Thank you for this clarification. Thus, we added 2 more tables to clarify the need of these rules and regulations in TN.

Reviewer 2 Report

This manuscript describes a combination of a rapid literature review and policy analysis with an end goal of developing policy drafts to improve access to methadone treatment in Tennessee. The topic is of high importance and the research was conducted in a methodologically rigorous way. There are areas for clarification that would further enhance the presentation of the manuscript that are noted below. Additionally, minor grammar and writing style edits are needed. I have noted some of these in my comments, but the authors are advised to apply the spirit of these edits to the entire manuscript. 

Abstract

1) Line 18: Should “naloxone” be “naltrexone”?

2) Line 20: What are “medication units”?

Introduction

3) Lines 83-84: Should be “OUD effectiveness” rather than “effectiveness OUD”.

4) Lines 114-119: What are the benefits/drawbacks of medication units compared to OTPs? This is central for establishing the need for this study. 

5) Lines 123-125: Wording of the sentence makes it difficult to follow. I believe it’s trying to state that fentanyl played a role in almost 90% of overdose deaths in Tennessee, which has led to an increase in methadone treatment demand.

6) Lines 132-133: Don’t need to state that OTPs represent methadone dispensing here as it is detailed more in the Methods.

Methods

7) Lines 181-182: Should be moved after treatment need is first mentioned in lines 177-178.

8) Line 189: “For” should be “from”.

Results

9) Line 243: Aren’t incarcerated individuals considered a “special population”?

10) Lines 274-277: Were there any findings from this study? The study focus is mentioned but not any results.

11) Lines 316-321: I’m not sure this is completely important, but if the authors feel this might be valuable, were there any counties that had an FQHC, pharmacy, and OTP?

12) Lines 327-329: How important is it to point out these differences if they were not statistically significant?

13) Lines 340-342: Were these differences tested statistically?

Discussion

14) Lines 420 and 421: “Country” should be “county”.

15) Line 430: Should be “would have access” not “has access”.

16) Policy Mapping: Consider which of this content should be provided in Results (i.e. new information should not be provided here, only implications of the results that have already been presented), and if information about Tennessee’s current policies should be in the Results as well, or potentially even in the Methods (in a “study setting” type of sub-section).

17) Lines 457-458: What are the “zoning restrictions” mentioned here?

18) Lines 460-470: Consider if there is value to readers in providing the language of the draft policy here as a table or appendix.

Author Response

This manuscript describes a combination of a rapid literature review and policy analysis with an end goal of developing policy drafts to improve access to methadone treatment in Tennessee. The topic is of high importance and the research was conducted in a methodologically rigorous way. There are areas for clarification that would further enhance the presentation of the manuscript that are noted below. Additionally, minor grammar and writing style edits are needed. I have noted some of these in my comments, but the authors are advised to apply the spirit of these edits to the entire manuscript. 

Abstract

1) Line 18: Should “naloxone” be “naltrexone”?

Response: Thank you for this clarification.  We amended the entire manuscript.

2) Line 20: What are “medication units”?

Response: Thank you for this clarification. As suggested by the other reviewers, we added a definition of the “medication units” in the introduction.

Introduction

3) Lines 83-84: Should be “OUD effectiveness” rather than “effectiveness OUD”.

Response: Thank you for the suggestion and sentence was restructured.

4) Lines 114-119: What are the benefits/drawbacks of medication units compared to OTPs? This is central for establishing the need for this study. 

Response: A sentence addressing the advantage of medication units was added.

5) Lines 123-125: Wording of the sentence makes it difficult to follow. I believe it’s trying to state that fentanyl played a role in almost 90% of overdose deaths in Tennessee, which has led to an increase in methadone treatment demand.

Response: Thank you for the suggestion and sentence was restructured

6) Lines 132-133: Don’t need to state that OTPs rep

resent methadone dispensing here as it is detailed more in the Methods.

Response: Thank you for the suggestion.

Methods

7) Lines 181-182: Should be moved after treatment need is first mentioned in lines 177-178.

8) Line 189: “For” should be “from”.

Response: Thank you for the recommendation and correction. 

Results

9) Line 243: Aren’t incarcerated individuals considered a “special population”?

Response: Thank you for the suggestion and sentence was restructured

10) Lines 274-277: Were there any findings from this study? The study focus is mentioned but not any results.

 Response: Thank you, this sentence was clarified.

11) Lines 316-321: I’m not sure this is completely important, but if the authors feel this might be valuable, were there any counties that had an FQHC, pharmacy, and OTP?

Response: Thank you for the recommendation.

12) Lines 327-329: How important is it to point out these differences if they were not statistically significant?

Response: Thank you for pointing this out, rationale for inclusion was added to the section.

13) Lines 340-342: Were these differences tested statistically?

Response: Thank you for this clarification. Unfortunately, there was not enough power to do additional statistical tests.

Discussion

14) Lines 420 and 421: “Country” should be “county”.

Response: Thank you for the correction. The text was amended.

15) Line 430: Should be “would have access” not “has access”.

Response: Thank you for the correction. The text was revised.

16) Policy Mapping: Consider which of this content should be provided in Results (i.e. new information should not be provided here, only implications of the results that have already been presented), and if information about Tennessee’s current policies should be in the Results as well, or potentially even in the Methods (in a “study setting” type of sub-section).

Response: Thank you for the recommendation. The text was amended and 2 tables were added that clarify the study.

17) Lines 457-458: What are the “zoning restrictions” mentioned here?

Response: Thank you for bringing to our attention. The text was amended.

18) Lines 460-470: Consider if there is value to readers in providing the language of the draft policy here as a table or appendix.

Response: Thank you for this valuable suggestion. We added 2 tables that strengthen our manuscript. Thank you again for this recommendation.

Round 2

Reviewer 1 Report

Authors have been responsive and cleared up some confusing issues with the prior manuscript.  There remain a few additional points that would improve the manuscript.  The most important is that they say a protocol was drawn up for the state to implement medication units run by pharmacies or FQHCs to address the underserved geographic areas of the state.  Since the policy draft itself is not part of the manuscript, I suggest the aim be revised to indicate that the study was designed to inform the drafting of a new public regulation or law to create medication units affiliated with already licensed OTP programs in the state.  

There remain some additional confusing lines. 

Line 376 refers to 20 county SES data being analyzed.  It is not clear why only those 20 counties and overall this paragraph does not add anything to the overall argument for the need to expand services since the data presented on income etc is mixed as to low SES.  I suggest eliminating it. 

Line 402 indicates of 55 high need counties, 76% need is met but its not clear if the authors mean 76% of the high need counties have an OTP.  That means 13 high need counties do not have a OTP presumably, and a majority of those (61%) are rural.  Rewriting lines 398-404 for clarity would be helpful. 

Line 482 refers to 16% of counties having direct access to an OTP, but it is not clear what portion of the population this represents. In addition, direct access is misleading depending on the size of the counties and the availability of public transportation.   Having one OTP per county would seem to be a minimum way to meet needs.  This is an important metric to consider.   Also line 487 refers to the figure 0.25%. Is it supposed to be 25%?? Is the point to say that capacity of OTPs has not changed hardly at all in response to more OUD?  

Author Response

Authors have been responsive and cleared up some confusing issues with the prior manuscript.  There remain a few additional points that would improve the manuscript.  The most important is that they say a protocol was drawn up for the state to implement medication units run by pharmacies or FQHCs to address the underserved geographic areas of the state.  Since the policy draft itself is not part of the manuscript, I suggest the aim be revised to indicate that the study was designed to inform the drafting of a new public regulation or law to create medication units affiliated with already licensed OTP programs in the state.  

Response: Thank you for the recommendation. The manuscript has been revised to state that it informed the policy draft. The authors had decided not to include the draft of the policy itself since it is currently undergoing departmental and legal review.

There remain some additional confusing lines. 

Line 376 refers to 20 county SES data being analyzed.  It is not clear why only those 20 counties and overall this paragraph does not add anything to the overall argument for the need to expand services since the data presented on income etc is mixed as to low SES.  I suggest eliminating it. 

Response: Thank you for the suggestion. The section was removed. It was originally added for completeness because the rapid review results suggested that SES might play a role in access/availability. The counties were chosen based on readily available data provided by the American Community Survey.

Line 402 indicates of 55 high need counties, 76% need is met but its not clear if the authors mean 76% of the high need counties have an OTP.  That means 13 high need counties do not have a OTP presumably, and a majority of those (61%) are rural.  Rewriting lines 398-404 for clarity would be helpful. 

Response: Thank you for pointing out the lack of clarity. This paragraph was revised to clarify if high or low need counites were being discussed.

Line 482 refers to 16% of counties having direct access to an OTP, but it is not clear what portion of the population this represents. In addition, direct access is misleading depending on the size of the counties and the availability of public transportation.   Having one OTP per county would seem to be a minimum way to meet needs.  This is an important metric to consider.   Also line 487 refers to the figure 0.25%. Is it supposed to be 25%?? Is the point to say that capacity of OTPs has not changed hardly at all in response to more OUD?  

Response: Thank you for your comment.

 The percentage of population was added.

The term “directly” was removed and replaced with in-county methadone dispensing to better represent that these individuals do not travel outside their county. The authors recognize transportation is a useful metric to consider when determining accessibility. Since transportation was not assessed, the authors focused on availability rather than accessibility. This is discussed in the limitation section of this study.

Line 487 (now Line 489) was revised to clarify that OTPs are nearing capacity which could limit access to methadone..

The paper focuses on availability rather than accessibility which is why transportation, travel time, and distance were not assessed.

The term directly was used to demonstrate that these individuals do not need to leave their county to have methadone dispensing available to them as we did not account for use and access of transportation in our review, which would be a useful metric to determine accessibility. The focus was placed on availability rather than accessibility since presence of an OTP is the minimum.